# Valencene, Nootkatone and Their Liposomal Nanoformulations as Potential Inhibitors of NorA, Tet(K), MsrA, and MepA Efflux Pumps in *Staphylococcus aureus* Strains

**DOI:** 10.3390/pharmaceutics15102400

**Published:** 2023-09-28

**Authors:** Cícera Datiane de Morais Oliveira-Tintino, Jorge Ederson Gonçalves Santana, Gabriel Gonçalves Alencar, Gustavo Miguel Siqueira, Sheila Alves Gonçalves, Saulo Relison Tintino, Irwin Rose Alencar de Menezes, João Pedro Viana Rodrigues, Vanessa Barbosa Pinheiro Gonçalves, Roberto Nicolete, Jaime Ribeiro-Filho, Teresinha Gonçalves da Silva, Henrique Douglas Melo Coutinho

**Affiliations:** 1Department of Biological Chemistry, Regional University of Cariri (URCA), Crato 63105-010, CE, Brazil; datianemorais@hotmail.com (C.D.d.M.O.-T.); gabriel.goncalves101@urca.br (G.G.A.); gustavo.miguelsiqueira@urca.br (G.M.S.); sheila.alves@urca.br (S.A.G.); saulorelison@gmail.com (S.R.T.); irwin.alencar@urca.br (I.R.A.d.M.); 2Department of Antibiotics, Federal University of Pernambuco (UFPE), Recife 50670-901, PE, Brazil; edersantana22@hotmail.com (J.E.G.S.); teresinha100@gmail.com (T.G.d.S.); 3Oswaldo Cruz Foundation (Fiocruz Ceará), Eusébio 61773-270, CE, Brazil; jpedroviana@alu.ufc.br (J.P.V.R.); pinheiro.vanessaf@gmail.com (V.B.P.G.); rnicolete@gmail.com (R.N.)

**Keywords:** bacterial resistance, efflux pumps, liposomes, nootkatone, valencene

## Abstract

Valencene and nootkatone are aromatic sesquiterpenes with known biological activities, such as antimicrobial, antioxidant, anti-inflammatory, and antitumor. Given the evidence that encapsulation into nanosystems, such as liposomes, could improve the properties of several compounds, the present study aimed to evaluate the activity of these sesquiterpenes in their isolated state or in liposomal formulations against strains of *Staphylococcus aureus* carrying efflux pumps. The broth microdilution method evaluated the antibiotic-enhancing activity associated with antibiotics and ethidium bromide (EtBr). The minimum inhibitory concentration was assessed in strains of *S. aureus* 1199B, IS-58, and RN4220, which carry the efflux proteins NorA, Tet(K), and MsrA. In tests with strain 1199B, valencene reduced the MIC of norfloxacin and EtBr by 50%, while the liposomal formulation of this compound did not show a significant effect. Regarding the strain IS-58, valencene, and its nanoformulation reduced norfloxacin MIC by 60.3% and 50%, respectively. In the non-liposomal form, the sesquiterpene reduced the MIC of EtBr by 90%. Against the RN4220 strain, valencene reduced the MIC of the antibiotic and EtBr by 99% and 93.7%, respectively. Nootkatone and its nanoformulation showed significant activity against the 1199B strain, reducing the EtBr MIC by 21.9%. Against the IS-58 strain, isolated nootkatone reduced the EtBr MIC by 20%. The results indicate that valencene and nootkatone potentiate the action of antibiotics and efflux inhibitors in strains carrying NorA, Tet(K), and MsrA proteins, which suggests that these sesquiterpenes act as efflux pump inhibitors in *S. aureus*. Therefore, further studies are needed to assess the impact of incorporation into liposomes on the activity of these compounds in vivo.

## 1. Introduction

Bacterial infections by antibiotic-resistant strains have become increasingly frequent, leading to high morbidity and mortality rates, especially in hospital settings, where patients usually acquire multidrug-resistant strains [1,2]. In this context, *Staphylococcus aureus* stands out as a potentially pathogenic bacterium. Like other species of the genus *Staphylococcus*, *S. aureus* primarily causes skin infections. However, due to the development of multiple resistant mechanisms, this pathogen can cause severe disease and death [3].

*S. aureus* is an opportunistic pathogen in humans and mammals, colonizing the skin and nasal mucosa without causing disease in most animals. On the other hand, this pathogen can also cause a vast number of conditions, especially when it encounters wounds and ruptures in the skin, among which the following stand out: folliculitis, impetigo, boils, abscesses, septic cellulitis, osteomyelitis, systemic infections, such as pneumonia, endocarditis, and bacteremia, in addition to other secondary infections, underlying inflammatory diseases such as dermatitis and arthritis [4].

The mechanisms of antibiotic resistance in *S. aureus* mainly involve the modification of the antibiotic target, the inactivation of the antibiotic, and the reduction of the intracellular concentration of the antibiotic in the bacterial cell [5]. In this sense, mutations in the pharmacological target are due to conformational changes that prevent antibiotic binding; the inactivation or structural modification of the antibiotic occurs through bacterial enzymes that cause the addition of chemical groups or direct degradation of the antibiotic [5]. On the other hand, the reduction of the intracellular concentration of the antibiotic may occur due to changes in the cell wall and negative regulation of porin expression, which hinders the entry of porin-dependent antibiotics [6]. However, efflux pump expression is one of the primary mechanisms associated with the decreased intracellular concentration of antibiotics. These proteins are present in Gram-positive and Gram-negative bacteria and have a remarkable capacity to expel antibiotics from the interior of the bacterial cell [7].

NorA, Tet(K), MsrA, and MepA, identified in resistant strains of *S. aureus*, are efflux pumps that extrude the antibiotics norfloxacin, tetracycline, erythromycin, and ciprofloxacin, respectively. These proteins can also transport other substrates, such as DNA intercalators and biocides, and therefore are associated with multidrug resistance [7]. It is known that many efflux pumps can be intrinsic to the bacterial genome or acquired through transposable elements such as plasmids, both from bacteria of the same genus and from different genera [8].

Therefore, one of the main strategies for controlling infections by bacteria carrying these proteins is the search for efflux pump inhibitors. These inhibitors can restore the activity and effectiveness of antibiotics against resistant bacteria. At the molecular level, efflux pump inhibitors can act by directly binding the pumps, uncoupling the proton gradient, inhibiting protein gene expression, or altering the fluidity of the plasmatic membrane [9,10]. Natural product research has demonstrated the effectiveness of natural products against resistant bacteria. Importantly, sesquiterpenes have been highlighted as efflux pump inhibitors and, therefore, are promising candidates in drug discovery [11,12].

Valencene is a highly unsaturated aromatic compound formed by two benzene rings. Evidence demonstrates that the high level of unsaturation favors its reactivity and is directly related to the compound bioactivity (Figure 1). In this context, valencene has aroused the scientific community’s interest due to its biological activities, such as anti-inflammatory, antitumor, antioxidant, antifungal, and antibacterial. However, the activity of valencene against *S. aureus* strains remains to be investigated [12,13,14].

Nootkatone is a sesquiterpene composed of three isoprene units (Figure 1). In addition to their aromatic properties, these sesquiterpenes have proven biological properties, such as anti-inflammatory, anticancer, antibacterial, hepatoprotective, neuroprotective, and cardioprotective. In addition, studies demonstrate that this compound is effective against Gram-positive bacteria, in addition to demonstrating promising effects against bacterial resistance mechanisms [12,15,16,17].

Studies by ours and other research groups have shown that the association of different compounds to nanoparticles can considerably improve their properties in various disease models. In this sense, drug encapsulation in liposomes is widely used to increase bioavailability, improve efficacy, and reduce side effects of therapeutic drugs. Indeed, liposomes can enhance the solubility and bioavailability of medicinal substances by protecting them from enzymatic degradation and reducing their toxicity in healthy tissues [18,19,20].

The investigation of different strategies for inhibiting efflux pumps is crucial for the identification of therapeutic targets, as well as for the discovery of adjuvants capable of enhancing antibiotic activity. Thus, considering the antibacterial potential of sesquiterpenes and the benefits liposomal formulations in drug development, the present study aimed to evaluate the effects of valencene, nootkatone and their liposomal formulations against *S. aureus* strains carrying efflux pumps (NorA, Tet(K), MsrA, and MepA).

## 2. Materials and Methods

### 2.1. Drugs and Reagents Used in the Synthesis of Liposomes

For the synthesis of liposomes, the lipids 1,2-dipalmitoyl-sn-glycero-3-phosphocholine, cholesterol, and dis-tearoylphosphatidylcholine were acquired from Sigma-Aldrich Co., Ltd. (St. Louis, MO, USA). Ethanol was used as the solvent for the lipids. The sesquiterpenes valencene and nootkatone were also obtained from Sigma-Aldrich.

### 2.2. Drugs and Reagents Used in Microbiological Tests

The antibiotics norfloxacin, tetracycline, and erythromycin were used as controls and specific substrates for strains 1199B, IS-58, and RN4220. The DNA intercalator ethidium bromide (EtBr) was used as a nonspecific substrate for these efflux pumps. 3-Chlorophenylhydrazone carbonylcyanide (CCCP) was used as a positive control and standard efflux pump inhibitor. These substances were obtained from Sigma-Aldrich Co., Ltd. Antibiotics, and sesquiterpenes were dissolved in 10% dimethylsulfoxide (DMSO) and subsequently diluted in sterile water to a concentration of 1024 μg/mL. CCCP was dissolved in a methanol/water (1:1 *v*/*v*) solution. EtBr was dissolved in sterile distilled water (1024 μg/mL), stored at −20 °C, and protected from light. Bacterial subculture and seeding were performed in Heart Infusion Agar (HIA, Difco, Forn El Chebbak, Lebanon), while Broth Heart Infusion (BHI) was used during the experiments.

### 2.3. Synthesis of Liposomes

Liposome nanoparticles containing sesquiterpenes were prepared by microfluidics. Briefly, a super-concentrated solution (50 mg/mL) of valencene and nootkatone was prepared for encapsulation. The organic phase of the nanoparticles was prepared from a lipid solution composed of 1,2-Dipalmitoyl-sn-glycero-3-phosphocholine (DPPC), cholesterol (CHOL), and Distearoylphosphatidylcholine (DSCP) in a proportion of 52:45:3 (mol/mol/mol), resulting in a final lipid concentration of 35 mM. After being weighed, the lipids were solubilized in absolute ethyl alcohol. The nanoparticles were prepared through the microfluidics technique, using the NanoAssemblr Benchtop equipment (Precision Nanosystems^TM^, Vancouver, BC, Cananda). In order to optimize encapsulation efficiency, the procedure was performed with a flow rate ratio of 2:1 and total flow ratio of 12 mL/min, considering the left and right syringes, respectively. The left syringe was filled with the aqueous phase, consisting of the diluted sesquiterpene solution, in a total volume of 3 mL, working volume of 3 mL, and flow rate of 8 mL/min. The right syringe, containing the lipid solution, was set with a total working volume of 1 mL and flow rate of 4 mL/min. In order to minimize losses, the volumes of the initial and final purges were adjusted to 100 µL. For the preparation of the control formulations (control liposomes) the left syringe was filled with PBS solution. After preparing the nanoparticles, the formulation was inserted into Amicon^®^ Ultra-15 3000 MWCO (Merck, Darmstadt, Germany) and centrifuged at 3000 rpm, 20 °C for 30 min to remove the residual solvent used in lipid solubilization. For this purpose, the samples were washed 10× in PBS (pH 7.2). This wash was the initial volume of the solution. The volume that did not pass through the membrane was recovered and completed to the expected volume using PBS buffer [21].

### 2.4. Bacterial Strains 

This study used strains of *S. aureus* expressing efflux pumps associated with antibiotic resistance. The strain 1199B expresses the NorA efflux pump, conferring resistance to fluoroquinolones such as norfloxacin; while the IS-58 strain expresses the Tet(K) pump, extruding antibiotics such as tetracyclines; and the RN4220 strain, which expresses the MsrA protein, confers resistance to macrolides such as erythromycin. These strains were provided by Prof. Glenn Kaatz (Wayne State University School of Medicine, Detroit, MI, USA) and Prof. Simon Gibbons (University College London, London, UK). Before the experiments, each strain was seeded and grown overnight at 37 °C in HIA.

### 2.5. Antibacterial Activity Analysis

The direct antibacterial action of sesquiterpenes in their isolated or nanoencapsulated forms was verified by the direct reduction of the minimum inhibitory concentration. For this, the broth microdilution method was used, using 10% of bacterial inoculum suspended in saline solution at a concentration corresponding to the value of 0.5 on the McFarland scale. Microtubes were added with the inoculum and 900 μL of BHI culture medium in a final volume of 1 mL. A 100 μL aliquot of this solution was distributed in 96-well microtiter plates. Serial dilutions (1:1) with sesquiterpenes and antibiotics were used to reach concentrations ranging from 512 μg/mL to 0.5 μg/mL. Plates were incubated in a bacteriological oven at 37 °C for 24 h. Bacterial growth was evaluated by adding resazurin, where blue staining indicated no growth and red/pink staining indicated bacterial growth. The minimum inhibitory concentration (MIC) was considered as the lowest concentration at which there no bacterial growth was observed [22,23].

### 2.6. Efflux Pump Inhibition Assessment by MIC Reduction

The test solutions were prepared in tubes by adding the inoculum, BHI medium, and the prepared compounds in subinhibitory concentration (MIC/8). The solution was distributed into the wells of a 96-well plate, followed by antibiotic dilution or EtBr added to an initial concentration of 1024 μg/mL. The negative control consisted of wells added with either antibiotic or EtBr only. Antibiotics were chosen according to the resistance pattern of each strain, that is, norfloxacin, tetracycline, and erythromycin, for strains 1199B, IS-58, and SA RN4220, respectively. The positive control CCCP was used as a standard efflux pump inhibitor at a concentration equivalent to MIC/8. The plates were placed in a bacteriological oven, and all the following steps were performed as described in Section 2.4 [22,23].

### 2.7. Efflux Pump Inhibition Evaluation through EtBr Fluorescence Emission

Strains 1199B and K2068 were seeded on Heart Infusion Agar (HIA) 24 h before the experiments and kept in a bacteriological oven at 37 °C. The inoculum was prepared in PBS until obtaining 1.5 × 10^8^ colony forming units (CFU), corresponding to the 0.5 value on the McFarland scale. The 1199B inoculum was prepared in a test tube by adding valencene or nootkatone at 100 µg/mL. The K2068 strain inoculum was added with valencene or nootkatone at 50 µg/mL in another group. The CCCP group (50 µg/mL) was used as a positive control for both strains. The negative control consisted only of the inoculum. In all tubes, PBS was added until reaching a final volume of 1 mL. After 90 min of incubation, EtBr (100 µg/mL) was added to all groups except the growth control. The solutions were again incubated for 1 h, then centrifuged at 10,000 rpm for 2 min, and washed twice with interspersed centrifugation cycles to remove EtBr and any supernatant substance. After centrifugation, the pellet was homogenized in PBS and distributed in the wells on a microplate. The reading was performed at 10, 20, 30, 40, 50, 60, and 70 min, using a fluorescence microplate reader (Cytation 1, BioTek^®^, Winooski, VT, USA) and Gen5™ 3.22 Software, with excitation of 530 nm and wavelength of 590 nm. The assay was performed in triplicate [24,25,26].

### 2.8. Statistical Analysis 

Statistical significance was determined through One-way ANOVA, followed by Tukey post hoc test, and Two-way ANOVA, followed by Bonferroni’s post hoc test. These analyzes were performed using GraphPad Prism software, version 5.0. All assays were performed in triplicate.

## 3. Results and Discussion

### 3.1. Physical-Chemical Profile of Liposomes

The physical-chemical characterization of the liposomes consisted of determining their average size, intensity, Zeta potential, concentration, PDI, and encapsulation efficiency. These characteristics are shown in Table 1.

The average size of nanoparticles can influence the biological activity of these drug delivery systems since this characteristic influences the biodistribution, absorption, and therapeutic efficacy of nanoparticles in different tissues and target cells. Recent studies have shown that smaller nanoparticles, with an average size close to 100 nm, generally have more significant biological activity and therapeutic efficacy. However, it is essential to emphasize that the biological activity of drug-based nanosystems is determined not only by the particles’ size but also by other factors such as composition, surface charge, morphology, and release properties of the encapsulated active compound [27,28].

The signal intensity detected in the nanoparticle tracking analysis provides information about the number of particles present in the sample. Therefore, the detected signal intensity is related to the number of particles in the sample and the encapsulation efficiency of the charged agents inside the nanoparticles. When calculating encapsulation efficiency, signal strength is used as an indirect measure to estimate the amount of encapsulated agent relative to the total amount of particles. Notably, the detected signal intensity can be influenced by factors such as the type of nanoparticle, the preparation method, the sample concentration, and the experimental conditions [29]. In this study, the nanoparticles presented satisfactory values of signal intensity.

Zeta potential is a measure of the electrical charge of nanoparticles and, as such, indicates their colloidal stability. Negative or positive values indicate that the particles have a net electrical charge, contributing to their stability. Several factors, such as pH, ionic strength, suspension composition, and interfacial interactions, influence this parameter [30,31]. Since values farther from zero indicate greater stability, the liposomes obtained in the present study present satisfactory characteristics with Zeta potentials of −11.6 mV for the valencene nanoformulation and −19.3 mV for the nootkatone nanoformulation.

Regarding the concentration, the nanoformulation containing valencene presented 4.83 × 10^8^ particles/mL, while the nootkatone formulation presented 2.3 × 10^8^ particles/mL. In therapeutic applications, the concentration of nanoparticles can influence the dose and effectiveness of the treatment. A study investigating the influence of polymer nanoparticle concentration on cancer therapy demonstrated that a higher concentration of nanoparticles resulted in more significant inhibition of tumor growth, indicating a relationship between the concentration and the therapeutic effect [32]. Furthermore, the concentration of nanoparticles is related to their biodistribution and toxicity. Studies have shown that highly concentrated nanoparticles can excessively accumulate in specific organs or tissues, resulting in toxic effects such as oxidative stress and cellular damage [33].

Finally, encapsulation efficiency measures the amount of active compound encapsulated within lipid nanoparticles, so higher values indicate greater encapsulation efficiency. The present study verified that liposomes containing valencene or nootkatone presented an encapsulation efficiency of 72.4% and 76%, respectively, which is considered technically significant.

### 3.2. Assessment of the Antibacterial Activity of Sesquiterpenes and Nanoformulations

Valencene showed direct antibacterial action against *S. aureus* 1199B and RN4220 strain, with MICs of 512 µg/mL and 256 µg/mL, respectively. Nootkatone also had a clinically relevant antibacterial effect against strains 1199B, IS-58, and RN4220, with MIC values of 406.4 µg/mL, 101.6 µg/mL, and 203.2 µg/mL, respectively. However, valencene or nootkatone nanoformulations failed to show clinically effective MIC (MIC ≥ 1024 µg/mL) for all strains tested (Table 2).

### 3.3. Effects of Sesquiterpenes on Antibiotic MIC against Efflux Pump-Bearing Strains 

Experiments with *S. aureus* strain 1199B, which carries the NorA efflux pump, demonstrated that unconjugated valencene reduced the MIC of norfloxacin and EtBr by 50%. On the other hand, the liposomal formulation of this sesquiterpene did not reduce the MIC of the antibiotic or EtBr (Figure 2A). Against the IS-58 strain (Tet(K) efflux pump), isolated valencene and its nanoformulation reduced the MIC of tetracycline by 60.3% and 50%, respectively. On the other hand, while the unconjugated sesquiterpene reduced the MIC of EtBr against the same strain by 90%, its liposomal formulation did not show a significant modulating effect (Figure 2B). Tests with the RN4220 strain, carrier of the MsrA efflux pump, showed a reduction in the MIC of erythromycin and EtBr by 99% and 93.7%, respectively, when these antibiotics were associated with isolated valencene. Comparable values were observed with the standard inhibitor, CCCP. However, the liposomal form of this substance did not show a significant effect (Figure 2C), suggesting that, under these experimental conditions, encapsulation does not improve the impact of sesquiterpenes in terms of reducing the MIC of antibiotics against resistant strains of *S. aureus*.

Nootkatone sesquiterpene, both in the isolated and liposomal forms, showed significant results against the 1199B strain, reducing the MIC of EtBr by 20.6%. However, none of these forms showed significant effects when associated with the antibiotic norfloxacin (Figure 3A). In tests with the IS-58 strain, nootkatone reduced the EtBr MIC by 20% (Figure 3B). The same result was observed against the RN4220 strain, where the isolated sesquiterpene reduced EtBr MIC, while the nanoformulation did not show significant effects (Figure 3C).

These results suggest that valencene alone can potentiate the action of fluoroquinolones, tetracycline, and macrolides. These results also indicate a possible inhibitory activity on the NorA, Tet(K), and MsrA efflux pump by valencene. On the other hand, in its encapsulated form, valencene potentiates only the activity of tetracyclines.

Regarding nootkatone, the results point to possible inhibition of *S. aureus* efflux pumps by sesquiterpene alone, while for the nanoformulation, this effect was more evident with the NorA protein. Indeed, the loss of effectiveness of sesquiterpene against some strains after encapsulation may occur due to interactions between the lipid composition of the nanoformulation and *S. aureus* membrane lipids, such as phosphatidylglycerol, lysyl-phosphatidylglycerol, cardiolipin, and glycophospholipids. However, how this interaction affects the action of sesquiterpene needs to be further investigated [34,35,36,37].

Notably, the hypotheses raised from this test are based on evidence that substances capable of reducing the MIC of EtBr and antibiotics that serve as substrates for efflux pumps can act as efflux inhibitors, enhancing the action of antibiotics [38,39,40,41,42,43,44,45,46].

### 3.4. Efflux Pump Inhibition through EtBr Fluorescence Emission

Analysis of fluorescence emission by the *S. aureus* strain 1199B demonstrated that valencene and nootkatone (100 µg/mL) increased this parameter by 237% and 18.7%, respectively, compared to the negative control (inoculum + EtBr) (Figure 4A). In tests with *S. aureus* K2068, valencene and nootkatone (50 µg/mL) increased the fluorescence emission rate by 63.7% and 13.6%, respectively. A significant increase in fluorescence was also observed with the standard CCCP inhibitor, attesting to the reliability of the results (Figure 4B).

The increased fluorescence observed following the treatment with these compounds corroborates the previous results, suggesting inhibition of the NorA and MepA efflux pumps, increasing EtBr’s intracellular concentration. EtBr is a DNA intercalating agent that only emits high fluorescence when bound to genetic material. Therefore, the emission of fluorescence from EtBr in this assay is directly proportional to the inhibition of efflux pumps [24,47,48].

Studies have shown that essential oils containing valencene as a principal constituent have significant antimicrobial activity [49,50,51,52,53,54]. A study with the essential oil of the leaves and flowers of *Vitex negundo* showed that both had valencene in their composition. However, while the essential oil from the leaves was active against *S. aureus*, the flower oil was active against *P. aeruginosa* [49]. In disk diffusion tests, Brazilian propolis essential oil containing valencene showed antibacterial activity against *E. coli* strains, Streptococcus pyogenes 75194, and *S. aureus* [50].

Docking was performed to analyze the interaction between bioactive compounds from *Pinus roxburghii* and *Cedrus* deodara and bacterial virulence proteins. Valencene’s binding solid affinity was demonstrated so that sesquiterpene showed more significant interaction with *S. aureus*, with binding affinity ranging from 5.1 to 7.7 kcal/mol [51]. Other species whose antibacterial activity was associated with the presence of valencene include *Leucas molissima* [52], *Salvia tomentosa* [53], and *Piper nigrum*, highlighting the pharmacological potential of this terpene against a vast number of bacterial strains [54]. Nevertheless, few studies have evaluated the biological activity of isolated valencene, and, therefore, this study is a pioneer in assessing the effect of this compound against strains of *S. aureus* carrying efflux pumps.

Evidence has shown that nootkatone exhibits antibacterial activity against Gram-positive bacteria such as *S. aureus*, *Enterococcus faecalis*, *Listeria monocytogenes*, *Corynebacterium diphtheriae*, and *Bacillus cereus* [16]. This compound also showed bacteriostatic action against *S. aureus* SJTUF 20758 at 200 μg/mL; and bactericidal activity against *S. aureus* ATCC 25923 at 400 μg/mL. Furthermore, this sesquiterpene inhibited biofilm formation in *S. aureus* strains at a concentration of 50 μg/mL, reducing the production of exopolysaccharides and inhibiting the expression of biofilm-related genes [17].

These findings corroborate the present research, demonstrating that valencene and nootkatone inhibited bacterial growth. Of note, the present study is innovative in that it also presents nootkatone as a possible inhibitor of efflux pumps in *S. aureus*.

Interestingly, nootkatone and valencene are structurally related since, in many processes, nootkatone can be obtained by bioconverting valencene by adding one oxygen atom and removing two hydrogen atoms. This process can be carried out by enzymes found in different microorganisms [55,56]. Bioconversion is a process widely used to obtain nootkatone. Since valencene is much more abundant than nootkatone, bioconversion is an economically attractive process [56,57]. In this sense, the applications of nootkatone in the food, cosmetic, chemical, and pharmaceutical industries stand out due to its unique odor and biological properties [58].

Therefore, valencene and nootkatone are structurally similar molecules. Hence, the promising results indicate that the bicyclic chemical structure and its radicals are involved in the action against bacterial strains that carry NorA, Tet(K), MsrA, and efflux pumps MepA.

## 4. Conclusions

In conclusion, valencene and nootkatone are potential efflux pump inhibitors in resistant strains of *S. aureus* expressing NorA, Tet(K), MsrA, and MepA proteins. However, incorporating these compounds into liposomes did not improve the effectiveness of these compounds as antibiotic resistance modulators, which may be due to interactions between the lipids that make up the nanoformulation and the bacterial membrane lipids.

Therefore, new studies are needed to describe the molecular targets involved in the action of these organic compounds, as well as to develop new nanotechnology strategies to improve the antibacterial properties of these compounds.

## Figures and Tables

**Figure 1 pharmaceutics-15-02400-f001:**
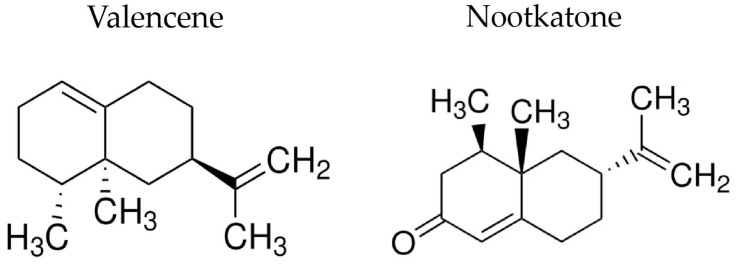
Chemical structures of valencene and nootkatone.

**Figure 2 pharmaceutics-15-02400-f002:**
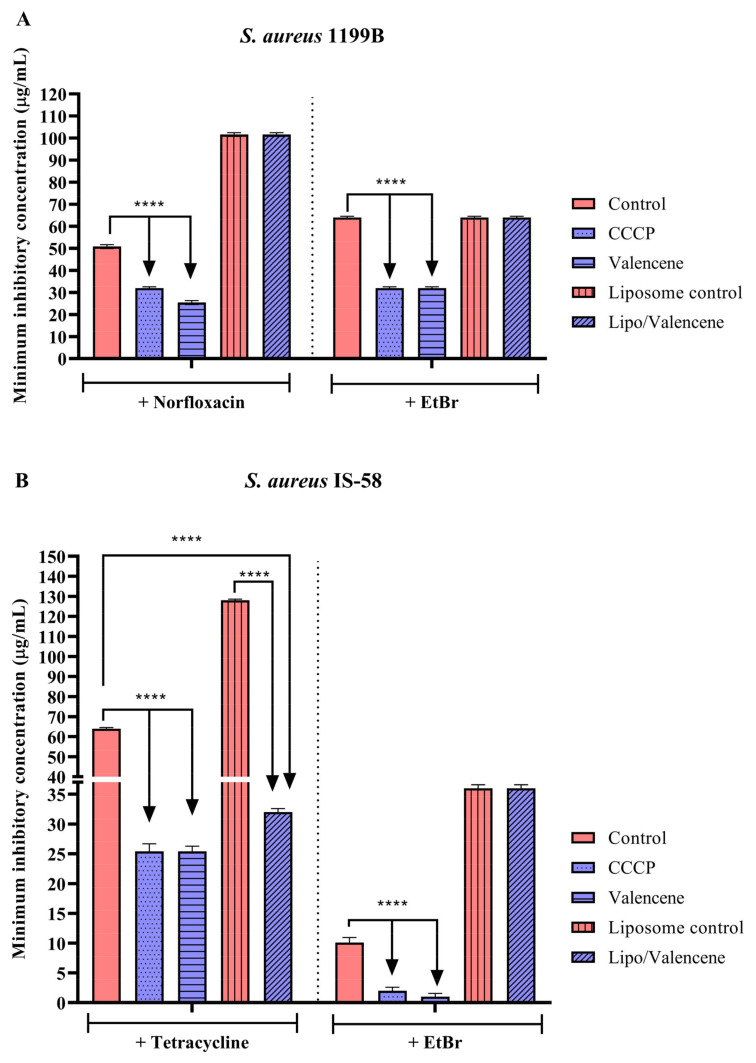
Evaluation of NorA, Tet(K) and MsrA inhibition by the valencene and its liposomal formulation associated with antibiotics and ethidium bromide against *S. aureus* strains 1199B (**A**), IS-58 (**B**), and RN4220 (**C**). CCCP = Carbonyl cyanide 3-chlorophenylhydrazone; EtBr = ethidium bromide; **** = *p* < 0.0001 vs. control was determined by Two-way ANOVA followed by Bonferroni post hoc.

**Figure 3 pharmaceutics-15-02400-f003:**
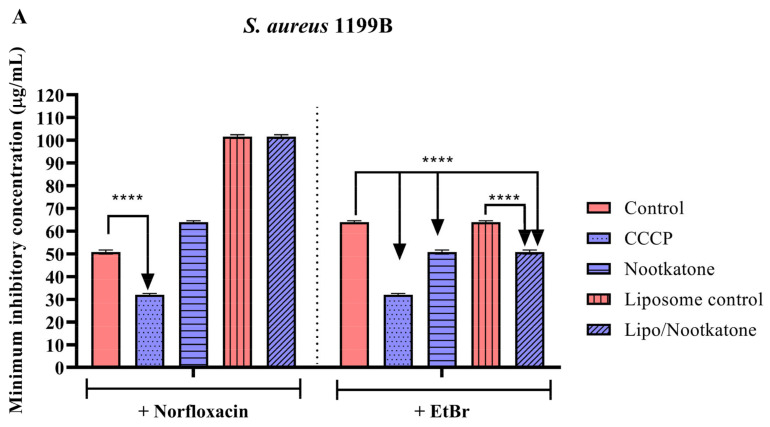
Evaluation of NorA, Tet(K) and MsrA inhibition by the nootkatone and its liposomal formulation associated with antibiotics and ethidium bromide against *S. aureus* strains 1199B (**A**), IS-58 (**B**), and RN4220 (**C**). CCCP = Carbonyl cyanide 3-chlorophenylhydrazone; EtBr = ethidium bromide; **** = *p* < 0.0001 vs. control was determined by Two-way ANOVA followed by Bonferroni post hoc.

**Figure 4 pharmaceutics-15-02400-f004:**
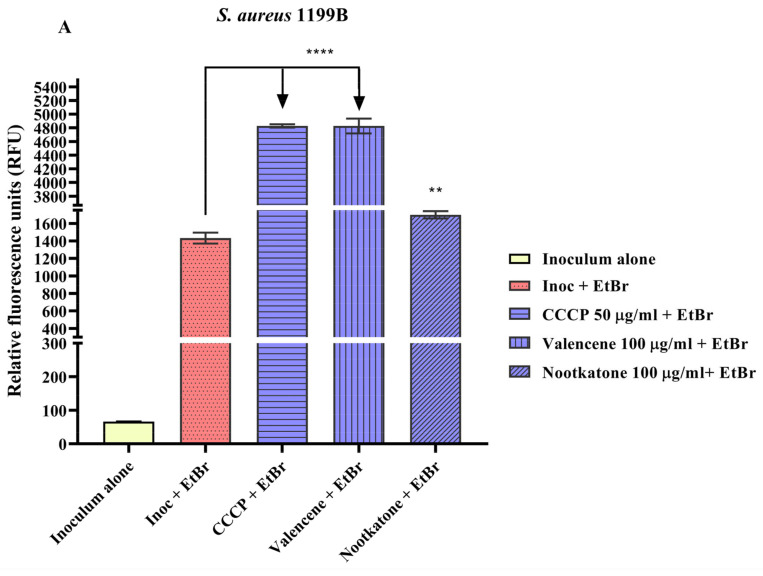
Evaluation of fluorescence emission and inhibition of the NorA and MepA efflux pumps in *S. aureus* 1199B (**A**), and *S. aureus* K2068 (**B**) treated with valencene and nootkatone. CCCP = Carbonyl cyanide 3-chlorophenylhydrazone; EtBr = ethidium bromide; Inoc = inoculum; **** = *p* < 0.0001 vs. Inoculum + EtBr; ** = *p* < 0.01 vs Inoculum + EtBr was obtained by analysis using One-way ANOVA followed by Bonferroni post hoc.

**Table 1 pharmaceutics-15-02400-t001:** Physicochemical characteristics of liposome nanoparticles containing valencene and nootkatone.

	Liposome/Valencene	Liposome/Nootkatone
Size	194.5 nm	147 nm
Intensity	5 a.u.	10 a.u.
Zeta potential	−11.6 mV	−19.3 mV
Concentration	4.83 × 108 particles/mL	2.3 × 108 particles/mL
Polydispersity index (PDI)	1.16	0.61
Encapsulation efficiency	72.4%	76%

**Table 2 pharmaceutics-15-02400-t002:** Antibacterial activity of valencene, nootkatone, and their liposomal nanoformulations.

	*S. aureus* 1199B	*S. aureus* IS-58	*S. aureus* RN4220
Control	128 µg/mL (Norfloxacin)	32 µg/mL (Tetracyclin)	32 µg/mL (Erythromycin)
Valencene	512 µg/mL*	≥1024 µg/ml	256 µg/mL *
Nootkatone	406.4 µg/mL *	101.6 µg/mL *	203.2 µg/mL *
Liposomal Valencene	≥1024 µg/mL	≥1024 µg/mL	≥1024 µg/mL
Liposomal Nootkatone	≥1024 µg/mL	≥1024 µg/mL	≥1024 µg/mL

* Clinically effective MIC values.

## Data Availability

All experimental data generated or analyzed during this study are included in the article.

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
