# Peer review of "Valencene, Nootkatone and Their Liposomal Nanoformulations as Potential Inhibitors of NorA, Tet(K), MsrA, and MepA Efflux Pumps in Staphylococcus aureus Strains"

_pharmaceutics, 2023, doi:10.3390/pharmaceutics15102400_

Round 1

Reviewer 1 Report

The manuscript titled “Valencene, nootkatone, and their liposomal nanoformulations inhibit NorA, Tet(K), MsrA, and MepA efflux pumps in S. aureus strains” written by Cícera Datiane de Morais Oliveira-Tintino focuses on the potentiating effects of the natural sesquiterpenes Valencene and nootkatone in association with Norfloxacin, Tetracyclin and Erythromycin in the treatment of S. aureus infections. Despite the manuscript reported valuable results, major revisions are required before its acceptance for the publication in Pharmaceutics.

Details:

-          The title should be modified since the results suggest a “possible” mechanism of action involving the inhibition of the efflux pumps, but no experimental evidence or specific assays confirmed this hypothesis. Moreover the liposomal nanoformulations showed no activit.

-          Introduction: The names of the bacterial strains must be in italics (e.g. lines 38, 40, 43…). The bibliography related to the antibiotic resistance in S. aureus needs an update, add the recent references Future Medicinal Chemistry, 2021, 13(6), pp. 529–531; Front. Cell. Infect. Microbiol., Sec. Clinical Microbiology, 10 – 2020, https://doi.org/10.3389/fcimb.2020.00107; Future Medicinal Chemistry, 2020, 12(5), pp. 357–359

-          Line 78: insert a figure with the chemical structures of Valencene and nootkatone

-          It is not clear why the authors decided to investigate the activity of Valencene and nootkatone against the efflux pumps. Clarify.

-          Pages 10, 11: check µg in fig 3

Author Response

Dear Reviewer, we appreciate your contribution. All the considerations have been acknowledged, and the corrections have been applied in order to improve our manuscript

Details:

Comment 1. The title should be modified since the results suggest a “possible” mechanism of action involving the inhibition of the efflux pumps, but no experimental evidence or specific assays confirmed this hypothesis. Moreover the liposomal nanoformulations showed no activit.

R1. Thank you for the suggestion. The new title is: Valencene, Nootkatone and their liposomal nanoformulations as potential inhibitors of NorA, Tet(K), MsrA, and MepA efflux pumps in Staphylococcus aureus strains

Comment 2-Introduction: The names of the bacterial strains must be in italics (e.g. lines 38, 40, 43…). The bibliography related to the antibiotic resistance in S. aureus needs an update, add the recent references Future Medicinal Chemistry, 2021, 13(6), pp. 529–531; Front. Cell. Infect. Microbiol., Sec. Clinical Microbiology, 10 – 2020, https://doi.org/10.3389/fcimb.2020.00107; Future Medicinal Chemistry, 2020, 12(5), pp. 357–359

R2: Corrected

Comment 3-Line 78: insert a figure with the chemical structures of Valencene and nootkatone.

R3: Done

Comment 4-It is not clear why the authors decided to investigate the activity of Valencene and nootkatone against the efflux pumps. Clarify.

R4: This is now clarified in the introduction.

Comment 5-Pages 10, 11: check µg in fig 3

R: Done

Reviewer 2 Report

Check the attached comments

Author Response

Response to Reviewer 2

Dear Reviewer, thank you for your contribution. All the considerations have been acknowledged, and the corrections have been applied.

Comment 1: Line 17: in liposomal formulations against strains of Staphylococcus aureus “here” after that add the (S. aureus) and used in remaining parts of manuscript instead complete name and italic them

R1. Done

Comment 2: Line 40: S. aureus primarily causes skin infections “here” S. aureus should be italic

R2: Thank you. Corrected

Line 43: S. aureus is an opportunistic pathogen in humans and mammals “here” S. aureus should be italic

R3. Thank you. Corrected

Comment 4. Lines 50: The mechanisms of antibiotic resistance in S. aureus “here” S. aureus should be italic

R4. Thank you. Corrected

Comment 5. Line 74-77: Rephrase the sentence “In the search for clinically promising efflux pump inhibitors, natural product research has identified aromatic compounds, such as sesquiterpenes, as new weapons in combating antibiotic resistance”

R5. The sentence has been rephrased: “Natural product research has demonstrated the effectiveness of natural products against resistant bacteria. Importantly, sesquiterpenes have been highlighted as efflux pump inhibitors and, therefore, are promising candidates in drug discovery”

Comment 6. Lines 173: PBS at a concentration of 1.5 x 108 colony-forming units (CFU) is it correct?

R6. A correction has been made.

Comment 7. Line 322-323: Studies have shown that essential oils containing valencene as a principal constituent have significant antimicrobial activity “here” add the reference

R7: Thank you for your observation. Done.

Comment 8. Line 332: interaction with S. aureus, with binding affinity ranging “here” S. aureus should be italic.

R8. Thank you. Corrected

Comment 9. Line 348: possible inhibitor of efflux pumps in S. aureus “here” S. aureus should be italic

R9. Thank you. Corrected

Comment 10. Line 384-537: Check the references following the reference's journal style

R10. Thank you. The reference style was corrected.

Reviewer 3 Report

It can be accepted in the current format.

Author Response

Thank you so much for your positive evaluation of our work.

Reviewer 4 Report

The research work evaluates the activity of Valencene and nootkatone in their isolated state as well as in liposomal formulations against strains of Staphylococcus aureus carrying efflux pumps. The results indicate positive action of antibiotics and efflux inhibitors in strains carrying NorA, Tet(K), and MsrA proteins. The comments are as follows,

1.       2. Materials and Methods: Please add materials as separate part.

2.       2.2. Drugs and reagents, please cite the materials selected, based on reagent or from standard protocols. 

To study the activity of Valencene and nootkatone in their isolated state or in liposomal formulations against strains of Staphylococcus aureus carrying efflux pumps.

Research is original and relevant to the delivery of bioactive. Liposomal formulation act most suitable to delivery this bioactive.  It may required further study for commercialization.

The study concluded that liposome are did not improve the effectiveness of these 364 compounds as antibiotic resistance modulators,

There no direct relicense of this work with published work. Few work on valencene and nootkatone are there with lipid based formulation, however this is totally novel concept and can be consider as proof new concept.

The methodology are well explained and can be reproducible.

More study should be done with other lipid, to conclude that, liposomes did not improve the effectiveness of these 364 compounds as antibiotic resistance modulators,

Author Response

Response to Reviewer 4

Dear Reviewer, we appreciate your contribution. All the considerations have been acknowledged, and the corrections have been applied.

Comment 1.       2. Materials and Methods: Please add materials as separate part.

 R1: Done

Comment 2.       2.2. Drugs and reagents, please cite the materials selected, based on reagent or from standard protocols. 

 R2: Thank You. The changes were performed

Comments “To study the activity of Valencene and nootkatone in their isolated state or in liposomal formulations against strains of Staphylococcus aureus carrying efflux pumps.

Research is original and relevant to the delivery of bioactive. Liposomal formulation act most suitable to delivery this bioactive.  It may required further study for commercialization.

The study concluded that liposome are did not improve the effectiveness of these 364 compounds as antibiotic resistance modulators,

There no direct relicense of this work with published work. Few work on valencene and nootkatone are there with lipid based formulation, however this is totally novel concept and can be consider as proof new concept.

The methodology are well explained and can be reproducible.

More study should be done with other lipid, to conclude that, liposomes did not improve the effectiveness of these 364 compounds as antibiotic resistance modulators,

R3.:  Thank you for your comments. All suggestions were applied to the manuscript.

Round 2

Reviewer 1 Report

The manuscript in the revised form can be accepted for the publication in the journal.